# Induction Chemotherapy for Locoregionally Advanced Sinonasal Squamous Cell Carcinoma and Sinonasal Undifferentiated Carcinoma: A Comprehensive Review

**DOI:** 10.3390/cancers15153798

**Published:** 2023-07-26

**Authors:** Katie L. Melder, Mathew Geltzeiler

**Affiliations:** Department of Otolaryngology—Head and Neck Surgery, Oregon Health & Science University, Portland, OR 97239, USA

**Keywords:** induction, chemotherapy, sinonasal malignancy, SNUC, SCC

## Abstract

**Simple Summary:**

Sinonasal malignancies are aggressive tumors that can be exceptionally difficult to treat. Surgery and adjuvant radiation therapy is considered first line therapy for both sinonasal squamous cell carcinoma (SNSCC) and sinonasal undifferentiated carcinoma (SNUC); however, the surgery for these lesions can cause significant morbidity and disfigurement. Induction chemotherapy has become a key part of multimodal therapy for patients with locoregionally advanced SNSCC and SNUC. In this review, we examine the literature surrounding induction chemotherapy, including its role in organ preservation and survival.

**Abstract:**

Sinonasal squamous cell carcinoma (SNSCC) and sinonasal undifferentiated carcinoma (SNUC) are two of the most common, high-grade malignancies of the sinonasal cavity. The standard of care for resectable lesions per the National Comprehensive Cancer Network (NCCN) guidelines includes surgical resection with negative margins plus adjuvant radiation therapy. However, surgery for locally advanced disease with both orbital and intracranial involvement is associated with significant morbidity and poor overall survival. Over the last decade, induction chemotherapy (IC) has emerged as part of a multimodal treatment strategy to optimize locoregional disease control and minimize substantial surgical morbidity such as orbital exenteration without compromising rates of overall survival. The response to IC both guides additional therapy and helps prognosticate a patient’s disease. This narrative review examines the data surrounding the management of patients with SNSCC and SNUC. The pros and cons of upfront surgical management plus adjuvant therapy will be explored, and the case for IC will be presented. The IC-specific regimens and treatment paradigms for SNSCC and SNUC will each be explored in detail. Organ preservation, treatment morbidity, and survival data will be presented, and evidence-based recommendations will be presented for the management of these patients.

## 1. Introduction

Malignant tumors of the sinonasal cavity are composed of varying pathologies with unique biological behavior. Sinonasal tumors are locally aggressive and often have invasion of surrounding structures (such as orbital or intracranial extension), resulting in advanced TNM staging at presentation [1,2]. Surgery is frequently the mainstay of treatment for resectable lesions; however, ultimately, histology drives the treatment paradigm. Sinonasal squamous cell carcinoma (SNSCC) and sinonasal undifferentiated carcinoma (SNUC) are two common and aggressive histologies that remain extremely difficult to treat, and over the past decade, the use of induction chemotherapy has been reintroduced as a valuable strategy for optimizing patient outcomes.

Most of the data for induction chemotherapy in sinonasal malignancies, specifically SNSCC, started within larger investigations of head and neck squamous cell carcinoma (SCC). The best evidence for organ preservation without oncologic outcome compromise with IC is in laryngeal SCC [3]. In the landmark VA study, cisplatin and fluorouracil (PF) were used as induction agents followed by radiotherapy. This allowed laryngeal preservation in 64% of this cohort. There are two phase three trials that looked at the addition of docetaxel as TPF (docetaxel, cisplatin, and fluorouracil) versus PF alone in patients with inoperable disease. In both trials, the dosages of cisplatin and fluorouracil were lower overall in the TPF group than the PF group. In the TAX 324 trial, patients underwent TPF or PF induction regimens followed by chemoradiation (CRT) with carboplatin. There was an OS benefit and improved locoregional control with TPF [4]. The TAX 323/EORTC 24971 trial randomized patients to either TPF or PF induction chemotherapy followed by radiotherapy alone [5]. Again, there was an OS benefit for the TPF group compared to the PF group, and progression free survival (PFS) also significantly improved. These two trials set the stage for TPF becoming the standard IC regimen in unresectable head and neck tumors. A meta-analysis of five randomized trials looking at TPF versus PF supported these findings and showed that the TPF group was associated with a significant reduction in progression, locoregional failure, and distant failure compared to PF [6]. Importantly, several previous studies with direct comparisons of induction chemotherapy regimens followed by CRT versus CRT alone in unresectable disease head and neck SCC have not shown a significant difference in overall survival [7,8,9,10]. Because of the increased toxicity of induction chemotherapy, IC fell out of favor, except for patients with advanced/T4b disease who felt healthy enough to tolerate the regimen. Over the last decade, however, IC has shown renewed interest for locally aggressive malignant sinonasal lesions.

SNSCC is the most common malignant tumor in the paranasal sinuses and represents 61% of all sinonasal malignancies [11]. According to the Surveillance, Epidemiology, and End Results (SEER) database, 5-year disease specific survival (DSS) is 23.4% [2]. SNUC is a largely undifferentiated tumor with some neuroendocrine differentiation and represents only 5% of sinonasal malignancies. Sinonasal neuroendocrine tumors (SNEC) and esthesioneuroblastoma (ENB) are in the same neuroendocrine family as SNUC and represent 2% and 4% of sinonasal tumors, respectively [11,12,13]. SNUC has a dismal prognosis, and the SEER database indicates a 5-year survival rate of 34.9% [14]. SNUC staging typically uses the American Joint Committee on Cancer or the Modified Kadish staging system. Comparatively, ENB has a high 5-year OS (73%); however, high Hyams grade tumors are associated with a much worse prognosis [15,16]. 

Surgical resection with negative margins followed by adjuvant therapy is the gold standard for treatment of the aforementioned sinonasal malignancies. Surgery has been shown to improve local control and overall survival (OS); however, in locally aggressive disease, surgical resection often has high morbidity and adverse functional outcomes. Positive margins have been associated with worse overall survival and disease-free survival (DFS) [17,18,19,20]. If an R0 resection is not achieved, surgery does not improve outcomes compared to nonsurgical treatment. Given their proximity to critical structures, negative margins may not be feasible or require craniofacial resection and/or orbital exenteration. Induction chemotherapy (IC) has therefore emerged as part of a multimodal treatment strategy. Given the rarity of these tumors, much of the literature to date is based on small, single-institution, retrospective studies which have inherent limitations and bias. There are two prospective trials; however, these also have very small numbers in each histopathologic subgroup. This review will discuss how IC can be used to not only prognosticate but also to help determine the ideal treatment course for patients with high-grade sinonasal malignancies. The IC approach in locally advanced disease allows a multi-disciplinary care team to provide the oncological optimal outcome while minimizing patient morbidity.

## 2. Psychological Burden and Complications of Surgical Treatment

Organ preservation is an integral objective of IC because of the significant morbidity associated with orbital exenteration, total maxillectomies, and craniofacial resections. Radical surgical resection in the head and neck often leaves patients with significant functional and body image concerns. Head and neck patients are at high risk of physiologic and societal dysfunction given the highly visible and socially significant part of their body undergoing treatment. In a study of 280 patients undergoing surgical treatment for head and neck cancer, 75% of patients reported body image concerns, such as embarrassment of losing an eye, and 33% of patients endorsed behavioral difficulties and social avoidance due to these concerns [21]. Additionally, a significant association was found between body image concerns and quality of life (QOL) domains involving physical, social, emotional, and functional wellbeing. These were tested using a body image scale and Functional Assessment of Cancer Therapy-Head and Neck Version (FACT-HN) patient reported outcome measures. 

Historically, most sinonasal malignancies were treated with open craniofacial resection. These procedures tended to be more morbid than current endoscopic techniques. Reported complications of open approaches in the literature are as high as 36%, with a majority of those being wound complications (infection, dehiscence, and flap necrosis) and central nervous system complications (CSF leak, meningitis, and pneumocephalus) [22]. Other serious but rare complications of open approaches include seizures and encephalomalacia. Psychiatric complications developed in 81% of patients undergoing craniofacial resection with orbital exenteration in one study measured by a psychiatric evaluation [23].

Current endoscopic surgery is the workhorse for sinonasal malignancies, with most pathologic subtypes showing similar or superior survival rates using this approach [24]. This approach provides a low-morbid option when the tumor biology and location are amenable [25]. Endoscopic endonasal approaches have improved many risks that were typically associated with open approaches; however, there are still significant complications that patients face, such as vascular and cranial nerve injuries. Patients who underwent endoscopic surgical resection have significantly better scores in physical function and emotional domains in the Anterior Skull Base Questionnaire (ASBQ) compared to open approaches [26]. Given the quality-of-life detriment with these extensive surgical procedures, orbital preservation should be attempted using multimodal treatments, specifically with induction chemotherapy.

## 3. Induction Chemotherapy in Sinonasal Squamous Cell Carcinoma

SNSCC is an aggressive tumor that often presents with advanced disease. Optimal treatment consists of multimodal therapy beginning with surgery when possible [27,28,29]. An expanding number of studies support the utilization of IC in T3/T4 SNSCC, especially for patients whose surgery would require orbital exenteration or craniofacial resection. For patients with orbital invasion, IC been shown in several retrospective studies to allow for organ preservation without any reduction in overall survival [30]. Responses to IC are typically graded based on the symptomatic and radiologic reduction of tumor volume with the following: complete response (CR), disappearance of all tumor; partial response (PR), >30% reduction in tumor burden; stable disease (SD); or progressive disease (PD), increase in ≥20% of tumor volume [31]. SNSCC is commonly treated with doublet or triplet therapy depending on the overall health of the patient and their presumed ability to handle more toxic regimens. The most common regimens include TPF or PF as shown by a systematic review by Khoury et al. of five SNSCC IC studies [32]. In their analysis, IC commonly included a platinum (94%), taxane (65%), and 5-fluorouracil (62%) agent. Cisplatin is often the chosen platinum agent; however, in patients with renal issues, carboplatin can be utilized as it is less nephrotoxic. Table 1 lists the major studies looking at IC in SNSCC in the past 20 years. 

SINART1 and SINART 2 studies are the only prospective data for induction chemotherapy in sinonasal lesions [33,34]. Both trials look at survival for various sinonasal pathologies, including SNSCC, SNUC, olfactory neuroblastoma, intestinal-type adenocarcinoma, and sinonasal neuroendocrine carcinoma. Patients were treated with up to five cycles of IC, and the regimen was selected based on pathology. In the SINART1 trial, which looked at resectable lesions, IC was followed by either curative CRT for those with a significant response (≥80% reduction of initial tumor diameter) or surgery with adjuvant RT/CRT. Overall, for all pathologies, with the majority being T4 (94%), 5-year OS was 46%, and PFS was 38%. SNSCC represents 37% of the resectable cohort, and they received TPF IC. The response rate was a PR of 54% and no CR. There was no progressive disease. Although this study is not powered for significance, it does show the feasibility and safety of IC followed by CRT. Overall, the regimen was well tolerated, with the most common toxicities including neutropenia, fatigue and diarrhea. One sudden cardiac death was reported but not thought to be related to treatment. The SINART2 trial looked at unresectable sinonasal lesions, and induction IC was followed with CRT or RT. The 5-year OS and 5-year PFS were 23.8% and 26.8%, respectively, for all pathologies. These lesions were all T4b. SNSCC represented 20% of unresectable pathologies, and these were treated with a TPF chemotherapy regimen. There was no partial or complete response, and disease progressed in 20% of patients. This did not show improved survival in SNSCC; however, this is a low-powered study, and limited conclusions can be made from this.

In the largest retrospective study published, 123 previously untreated patients with stage III/IV disease (89% T4) underwent a variety of IC regimens [35]. The most common regimen was TP (35.5%) and TPF (27.6%); however, several other agents were utilized, including ifosfamide and cetuximab. The majority of patients were responders with a CR or PR of 57%. This cohort had a high rate of orbital invasion (67%), yet only 18.5% of those with orbital invasion underwent exenteration showing successful organ preservation with IC. There is a stark dichotomy of 2-year OS in the PR/SD group (68.2%) and PD (33.3%) which is consistent with the other literature that states that responders have improved survival. Interestingly, in each response group (CR, PR, SD, and PD), there was no difference in 5-year OS in the definitive treatment groups (surgery, CRT, and combinations). This highlights the lack of a clear consensus in the literature on definitive treatment regimen following IC for SNSCC. Table 1 highlights the varied percentages for subsequent treatments. Noronha et al. evaluated 41 patients with advanced SCC (T4 disease in all) of maxillary sinus, and CRT was the most common definitive treatment utilized after IC (58.5%) [36]. A doublet regimen of TP was primarily used and showed a response rate of 61.9% with an OS of 35% at 3 years. 

Ock et al. investigated the role of induction chemotherapy for orbital preservation in a cohort of 21 patients, with the majority being T4 (80.9%) [37]. The most common chemotherapy regimen used was TPF (52.4%) followed by PF (38.1%). A partial response to chemotherapy was obtained in 61.9%, and of those that responded, orbital preservation was achieved in all patients. This highlights the evidence for IC in orbital preservation. Most of these patients went on to get CRT (53.9% for PR and 50% for SD/PD). At 5 years, OS in the PR group was significantly improved compared to the SD/PD group (65.8% and 25%, respectively, *p* = 0.036). 

Nyiriesy et al. looked at multiple sinonasal pathologies but performed a secondary analysis of SNSCC comparing those who were treated with and without IC [38]. As expected, those treated with IC were significantly more likely to have orbital, skull base, or dural involvement as well as a higher T stage. However, even with the IC cohort having significantly more aggressive and locally advanced disease compared with the non-IC cohort, there were no significant differences in two-year OS or DFS.

In a matched analysis of 26 patients who received either IC or upfront definitive therapy with primary surgery or CRT, Murr et al. found that the IC cohort demonstrated significantly improved 3-year OS (100% vs. 48.4%, *p* = 0.016) [27]. Weekly cetuximab, carboplatin, and paclitaxel was the most common regimen utilized. Overall, 84.6% responded to IC with either a CR or PR, and 23% experienced adverse events, including neutropenia, type I hypersensitivity, and acute kidney injury. Compared to the standard of care cohort, IC patients did not exhibit higher DFS; however, they demonstrated a significantly improved 3-year OS (100% vs. 48.4%, *p* = 0.016). Importantly, the authors noted that patients treated with IC had higher mean comorbidity scores, suggesting that their IC regimen may be well-tolerated, even in patients with multiple comorbidities.

Several other studies have also showed favorable responses to induction chemotherapy, including improved overall survival and an increased chance of organ preservation [30,39,40]. Turri-Zanoni et al. looked at 163 mixed sinonasal pathologies with orbital involvement [30]. Twenty-four percent were given IC, and for those with SNSCC and SNUC, this consisted of docetaxel, cisplatin, and 5-fluorouracil for a maximum of five cycles. In those that responded (CR/PR) to IC, 100% had orbital preservation at the end of the study, and 62% were down staged. Non-responders demonstrated only a 46% preservation rate, while those who were not given IC had a preservation rate of 72%. Orbital apex involvement translated to a dismal prognosis with a 5-year OS of 14.6%, regardless of treatment. 

Functional outcomes for orbital preservation are generally favorable. One study by Lisan et al. [41] shows an overall low rate of impairments which included 9% diplopia, 14% keratitis, and 11% epiphora. Of note, two patients presented post treatment with blindness (5%), of which both cases were secondary to rapid local recurrence in orbit. In the Turri-Zanoni study [30], the majority of patients who underwent orbital preservation had a functional eye without significant impairment (63%), 32.8% had a functional eye with some impairment, and a small minority (4%) had a nonfunctional eye. 

These studies highlight the promising role of IC in SNSCC and how response dictates prognosis. Table 1 highlights the high OS survival rate in this cohort, specifically when divided by IC responders. Unfortunately, there is no data on how to stratify patients based on who will most likely respond to IC before having patients undergo these toxic regimens. 

**Table 1 cancers-15-03798-t001:** Retrospective studies investigating induction chemotherapy in sinonasal squamous cell carcinoma.

Paper	n	T4 Disease	IC Regimen	IC Response	Subsequent Treatment	Orbital Preservation	OS
Abdelmeguid, 2021 [35]	123	110 (89.0)	TP—41 (35.5) TPF—34 (27.6) TP+I—26 (21.1) PF—10 (8.1) TP+Cetux—8 (6.5) P+I—2 (0.16) P+Gem—1 (0.8) P+Etop—1 (0.8)	CR—6 (5) PR—64 (52) SD—32 (26) PD—21 (17)	Sx—53 (44.2)	81.5% pts with OI preserved	2-year OS: PR/SD—68.2% PD—33.3%
Hanna, 2011 [39]	46	37 (80)	TF—9 (20) TP—14 (30) TPF—9 (20) TP+I—14 (30)	CR/PR—31 (67) SD—4 (9) PD—11 (24)	Sx—46 (52%)	NR	2-year OS: 77%
Hirakawa, 2016 [42]	43	32 (74.4)	PF—43 (100)	** Grade 2/3: 15 (34.9)	Sx—43 (100)	Overall: 55.8%	5-year OS: G0/1—54.3% G2/3—93.3%
Noronha, 2014 [36]	41	41 (100)	TP—34 (83) TPF—7 (17)	PR—16 (39) SD—18 (43.9) PD—7 (17.1)	Sx—12 (29.3) CRT—24 (58.5) RT—1 (2.4)	NR	3-year OS: 35%
Ock, 2016 [37]	21	17 (80.9)	PF—8 (38) TP—2 (10) TPF—11 (52)	PR—13 (61.9) SD—6 (28.6) PD—2 (9.5)	CRT—11 (52.4) RT—3 (14.3) Sx—6 (28.5)	Overall: 85.7%	5-year OS: PR—65.8% SD—25% PD—25%
Choi, 2013 [40]	17	9 (53)	PF—17 (100)	CR—1 (5.9) PR—6 (35.3) SD—4 (23.5) PD—6 (35.3)	CRT—5 (29) Sx/RT—12 (71)	NR	5-year OS: 40%

Values are presented as n (%); Cetux = Cetuximab; CR = complete response; CRT = chemoradiation therapy; Etop = Etoposide; F = 5-fluorouracil; Gem = Gemcitabine; I = ifosfamide; OI = orbital involvement; OS = overall survival; P = platinum; PD = progressive disease; PR = partial response; RT = radiotherapy; SD = stable disease; Sx = surgery; T = taxane; and V = vinblastine. ** Pathologic responses were utilized.

## 4. Induction Chemotherapy in Sinonasal Undifferentiated Carcinoma and Olfactory Neuroblastoma

SNUC is a rare, aggressive tumor that often presents at advanced stages with orbit or intracranial involvement in addition to having a propensity to metastasize. Multimodal treatments are almost always utilized with this pathology. In a meta-analysis of 390 patients with SNUC, double and triple modality treatment was significantly associated with improved survival over single modality [43]. IC has become increasingly recognized as the branchpoint of the treatment paradigm in advanced lesions. One institution reviewed their experience with SNUC and found that since 2016, patients were more likely to be managed with IC followed by CRT, which roughly mirrors the treatment of these patients in the field [44]. Like SNSCC, there are no randomized control trials, largely due to the small number of cases that each individual institution sees; however, there are several retrospective studies showing the benefit of IC which help formulate conventional treatment paradigms. 

In a landmark study by Amit et al., 95 patients with untreated SNUC were given doublet therapy with a platinum and either etoposide or docetaxel [45]. Cisplatin was used unless the patient had renal insufficiency, significant hearing loss, or peripheral neuropathy, in which case carboplatin was utilized. The median number of IC cycles administered was three; however, there was a range of up to five cycles. Most of these patients had T4 disease (69.5%). The percentage of patients who were responders with either a complete or partial response was 67.4%. In responders, 5-year DSS was 81% for those that went on to get definitive CRT, and only 54% of those people got surgery followed by adjuvant treatment. For non-responders, the DSS was abysmal in comparison; however, it was much worse for those that underwent CRT compared to surgery plus adjuvant therapy (0% and 39%, respectively). This paper shifted the paradigm to a clear data-driven pathway that shows that responders should get definitive CRT and non-responders should go on to surgery. This contrasts with the SNSCC data for treatment after response to IC, which is much less defined. 

Looking back at the SINART1 and SINART2 studies in relation to SNUC, we see more convincing data for the use of IC in SNUC when compared to SNSCC [33,34]. In the SNART1 trial, SNUC represented the majority (43%) of resectable lesions and was treated with TPF with a PR of 47% and CR in 13%. Responders represents the vast majority of patients with resectable SNUC. This aligns with other studies suggesting that SNUC has one of the best response rates to IC. There was progressive disease in only 13% of people. SNUC and neuroendocrine histotypes showed the highest objective response rate (ORR) in this cohort at 60% and 75%, respectively. In the SINART2 trial, SNUC represented 44% of cohort, and it was also treated with TPF chemotherapy. There was a 46% PR and an 18% CR, which is similar to that seen in resectable SNUC. No progressive disease was found. ONB represented 13% of the pathology and was treated with etoposide and cisplatin. There was 100% stable disease. There was no subtype analysis done on SNUC pathology.

In contrast, there are studies that do not show strong evidence for IC. One recent National Cancer Database study of SNUC patients by Lehrich et al. looked at the National Cancer Database from 2004–2015 and showed that IC did not affect OS compared to those that did not undergo IC prior to definitive treatment [46]. This contrasting end point may be secondary to the highly heterogenous IC agents and dosing utilized as well as the IC cohort having statistically significant overall health with more comorbidities. Interestingly, there were socioeconomic metrics that were associated with a significantly worse OS, namely having government insurance and living in a region where >13% of the population was without a high school diploma. This suggests a low health literacy rate and likely later diagnosis.

Another neuroendocrine tumor that has been investigated with IC is high-grade ENB. Su et al. looked at 15 patients with ENB with extensive local invasion and/or nodal disease who were treated with IC [47]. The most common regimen was cisplatin and etoposide (80%). Patients were imaged after two cycles of treatment, and responders were eligible for up to two additional rounds of IC. The overall rate of responders was 68% and was higher in the high Hyams grade group (78%). 

There is not a consensus on the optimal number of IC cycles or agents; however, as seen in Table 2, all studies used a platinum-based agent. Cisplatin and etoposide are common agents utilized in the most recent literature. Patil et al. reported 25 patients (13 SNUC and 12 ENB) where cisplatin and etoposide IC were also utilized and concluded that this improved outcomes (the 2-year OS was 78.5%) [48]. This contrasts with SNSCC where etoposide is rarely utilized.

With recent advances, histologic subtypes are beginning to play a more vital role in determining treatment and prognosis; however, this research is still in its infancy. Tumors that were initially classified as poorly differentiated/undifferentiated carcinomas can be further classified based on immunohistochemistry staining. There are several recently discovered distinct entities, such as SMARCA4 deficient sinonasal carcinomas and SMARCB1 (INI-1) deficient sinonasal carcinoma [49,50,51]. SMARCA4 deficient carcinoma is thought to have a neuroendocrine origin, while SMARCB1 deficient carcinoma represents an emerging poorly differentiated/undifferentiated sinonasal carcinoma. Current literature does not definitively say if SMARCB1 deficient carcinoma responds similar as SNUC to IC, and some of the best studies looking at IC and SNUC do not mention these subtypes [45]. 

The current literature supports the finding that IC responders have better outcomes and prognosticate a patient’s disease. Unfortunately, there is still no definitive way to know who will respond to IC and who will not. This research is ongoing, and a recent study by Takahashi et al. found 34 differentially expressed genes that distinguish responders from non-responders and 16 gene pairs which were associated with a response to induction therapy [52]. Future research is needed to confirm the sensitivity and specificity of these genes. 

SNUC is an aggressive tumor with a high propensity to metastasize. IC has been found to increase OS in responders and predict CRT responsiveness which therefore drives definitive treatment modality selection. 

## 5. Toxicity of Chemotherapeutic Agents

As demonstrated in Table 1 and Table 2, there can be a wide variety of chemotherapeutic agents used to treat sinonasal malignancies. It is imperative that we understand the toxicity of these agents for patient counseling. Using the Common Terminology Criteria for Adverse Events, the severity of reactions can be graded as follows: Grade 1, mild; Grade 2, moderate/local or noninvasive intervention indicated; Grade 3, severe or medically significant but not life threatening/hospitalization; Grade 4, life threatening consequences with urgent intervention needed; and Grade 5, death-related adverse event. 

Studies are extremely variable in reported side effects. In a SNSCC cohort who received an induction regimen of docetaxel, cisplatin, and fluorouracil followed by weekly cisplatin, Grade 3 nausea/vomiting and neutropenia was experienced by 14.3% and 9.5% of people, respectively [37]. There were no reported Grade 4 adverse events. Another study which used an induction regimen of a platinum-based doublet regimen coupled with etoposide or docetaxel reported a much higher rate of Grade 3 or 4 toxicities. Adverse hematologic effects were 34% (neutropenia 20%, thrombocytopenia 11%, and febrile neutropenia 3%), and non-hematologic effects were 26% (nausea/vomiting 18%, pulmonary embolism 3%, deterioration in renal function 3%, and acute myocardial infarction 2%). Hearing impairment was reported in 25% of the cohort [45].

There is a push to identify patients who are at a higher risk of developing toxicities. One study looking at 92 head and neck cancer patients receiving docetaxel, cisplatin, and 5-fluorouracil showed that low skeletal muscle mass is a predictive factor of toxicity to IC [53]. Another study of 113 patients treated with TPF had 6% toxic deaths, and there was an increased risk of death with preexisting liver dysfunction [54]. Ultimately, chemotherapeutic agents have a significant side effect profile. The toxicity for these agents remains one of the biggest issues, and there are attempts in the reported studies to improve toxicity by optimizing scheduling, agents, and dosing; however, no consensus exists. 

## 6. Conclusions

Induction chemotherapy has shown promising results as a part of multimodal treatment in patients with locally advanced sinonasal tumors, specifically for SNSCC and SNUC. The gold standard treatment remains upfront surgical resection followed by adjuvant radiation +/− chemotherapy; however, there are patients that are more suited to IC—especially those with locally advanced disease involving the orbit or intracranial structures. Tumors with a favorable response to induction chemotherapy are associated with improved outcomes and have increased rates of orbital preservation compared to non-responders. There is no consensus on the exact chemotherapeutic regimen; however, a platinum backbone is always utilized. Squamous cell carcinoma regimens tended to be augmented with a taxane, while neuroendocrine lesion doublets included etoposide. While there is toxicity associated with chemotherapeutic agents, many patients can tolerate two to three cycles. The literature supports IC followed by definitive local therapy for the management of advanced SNSCC, SNUC, and ENB for organ preservation and improving overall survival. These conclusions are based primarily on retrospective studies; therefore, there are inherent biases associated with that. Further studies including multi-institutional and prospective studies are required to validate these conclusions.

## 7. Future Directions

Larger, prospective trials will further inform which pathologic subtypes respond best to IC. A better understanding of the biology and genomics of sinonasal malignancies will improve targeted therapies. Further studies are working on the biologic markers for the IC responders as this is the key to optimizing survival while preventing unnecessary toxicities in patients that will respond and thus gain no benefit from IC regimen. 

## Figures and Tables

**Table 2 cancers-15-03798-t002:** Retrospective studies investigating induction chemotherapy in sinonasal undifferentiated carcinoma.

Paper	N, Path	Staging	IC Regimen	IC Response	Orbital Preservation	OS/DFS/PFS
Amit, 2019 [45]	95, SNUC	T4: 66 (69.5)	P+Etop or Docetaxel	CR/PR—64 (67.4) SD/PD—31 (32.6)	Overall: 96.8%	5-year DSS: PR/CR→CRT—81% PR/CR→Sx—54% <PR→CRT—0% <PR→Sx—39%
Patil, 2016 [48]	12, ENB; 13, SNEC	Grade III–IV: 23	P+Etop 25 (100)	CR/PR—20 (80) SD/PD—5 (20)	92.3% pts with OI preserved	2-year OS: 78.5%
Su, 2017 [47]	15, ENB	T4: 12 (80) KadishC: 12 (80) HyamsIV: 3 (20)	P+Etop—12 (80) Cycl+Doxorubicin +V: 2 (13)	CR—7 (46.7) PR—3 (20) SD—4 (27) PD—1 (6)	50% pts with OI preserved. Overall: 80%	5-year DFS/OS: 71%/78% 5-year DFS: CR: 100% PR/SD/PD: 46%

Values are presented as n (%); CR = complete response; CRT = chemoradiation; Cycl = Cyclophoshamide; DSS = disease specific survival; ENB = esthesioneuroblastoma; Etop = Etoposide; OI = orbital involvement; OS = overall survival; P = platinum; PD = progressive disease; PR = partial response; SD = stable disease; SNEC = sinonasal neuroendocrine carcinoma; SNUC = sinonasal undifferentiated carcinoma; Sx = surgery; and V = vincristine.

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
