# Peer review of "Induction Chemotherapy for Locoregionally Advanced Sinonasal Squamous Cell Carcinoma and Sinonasal Undifferentiated Carcinoma: A Comprehensive Review"

_cancers, 2023, doi:10.3390/cancers15153798_

Round 1
Reviewer 1 Report
Congratulations to both authors on a well-written manuscript. With a narrative review, the review of literature is unsystematically searched and data is minimally extracted to answer a broad question that may be vaguely stated. This manuscript is extremely informative and worthy of a publication if the authors would reconsider changing it to a scoping review with some meta analysis for overall survival (OS), disease free survival (DFS),.. Scoping review will identify the gaps that are missing in the literature which will help aid future studies.
Author Response
Thank you for your suggestions. All literature with a high volume that included induction chemotherapy regimens in SCCa and SNUC in the last 20 years was included in the paper and reviewed. There was one recently published paper SINART 1 and 2 since the draft of the manuscript that was added. We believe this adds to the body of literature and provides a strong, comprehensive review as is.
If this addition is not adequate, a scoping review can be considered. Thanks so much for your time.
Reviewer 2 Report
In this paper, the authors review the current evidence that supports the use of induction chemotherapy in sinonasal SCC and in SNUC. The article is well written and the references are appropriate ; the conclusion is supported by the data.
Three questions/suggestions:
- - Most of the available evidence in this field is based on retrospective studies with many potential biases ; this should be underlined very early in the paper, probably in the introduction and maybe in the abstract.
- - Could the authors elaborate on the importance of an accurate pathological diagnosis, especially for SNUC ? Many of the cited studies probably included different histological subtypes (SMARCB1/SMARCA4 deficient…) with potential differences in chemo/radiosensitivity and global prognosis
- - Could the authors include and discuss the results of two prospective studies that have been recently published on this topic (SINTART 1 and SINTART 2 – PMID 37164774 and 37163806) ?
Author Response
- Great point. The retrospective nature of most of the studies is indeed a weakness. This has been added.
- This has been added.
- Yes, I believe this was published just after the initial manuscript was completed. This has been added.
Thanks so much
Round 2
Reviewer 1 Report
Authors did a wonderful job in addressing all the comments by reviewers.